# Investigational medicinal products, related costs and hospital pharmacy services for investigator-initiated trials: A mixed-methods study

Ala Taji Heravi[1,2]*, Anne Henn[3], Stefanie Deuster[3], Stuart McLennan[4], Viktoria Gloy[1], Vera Ruth Mitter[5], Matthias Briel[1,6], for the MAking Randomized Trials Affordable (MARTA) Group[¶]

1 Department of Clinical Research, Basel Institute for Clinical Epidemiology and Biostatistics, University Hospital Basel and University of Basel, Basel, Switzerland, 2 Swiss Tropical and Public Health Institute, Basel, Switzerland, 3 Hospital Pharmacy, University Hospital Basel, Switzerland, 4 Institute of History and Ethics in Medicine, Technical University of Munich, Munich, Germany, 5 Department of Gynaecology, University Hospital Bern, University of Bern, Bern, Switzerland, 6 Department of Health Research Methods, Evidence, and Impact, McMaster University, Hamilton, Ontario, Canada

¶ Membership of the the MAking Randomized Trials Affordable (MARTA) Group is provided in the Acknowledgments.
* ala.tajiheravi@usb.ch

**Data Availability Statement:** All data generated or numerically analyzed during this study are included in this published article and its supplementary

## Abstract

### Background

Conducting high quality investigator-initiated trials (IITs) is challenging and costly. The costs of investigational medicinal products (IMPs) in IITs and the role of hospital pharmacies in the planning of IITs are unclear. We conducted a mixed-methods study to compare planned and actual costs of IMPs in Swiss IITs, to examine potential reasons for differences, and to gather stakeholder views about hospital services for IITs.

### Methods

We included all IITs with IMP services from the Basel hospital pharmacy invoiced between January 2014 and June 2020 (n = 24). We documented trial and IMP characteristics including planned and actual IMP costs. Our working definition for a substantial cost difference was that the actual IMP costs were more than 10% higher than the planned IMP costs in a trial. We conducted semi-structured interviews with investigators, clinical trials unit and hospital pharmacy staff, and qualitatively analyzed transcribed interviews.

### Results

For 13 IITs we observed no differences between planned and actual costs of IMPs (median, 11'000 US$; interquartile range [IQR], 8'882–16'302 US$), but for 11 IITs we found cost increases from a median of 11'000 US$ (IQR, 8'922–36'166 US$) to a median over 28'000 US$ (IQR, 13'004–49'777 US$). All multicenter trials and 10 of 11 IITs with patients experienced substantial cost differences. From the interviews we identified four main themes: 1)

information files. In order to access administrative trial files including invoices at the hospital pharmacy, we needed to sign confidentiality agreements allowing the publication of aggregated data only and ensuring that individual studies and principal investigators are not identifiable. Interview participants allowed us to use the transcripts for qualitative data analysis but not the dissemination or sharing of transcripts with others.

**Funding:** The authors received no specific funding for this project. VRM is supported by the Swiss National Science Foundation through an Early Postdoc. Mobility career grant (P2BEP3_191798).

**Competing interests:** The authors have declared that no competing interests exist.

**Abbreviations:** API, Active Pharmaceutical Ingredient; CTU, Clinical Trial Unit; GMP, Good Manufacturing Practice; IITs, Investigator-Initiated Trials; IMP, Investigational Medicinal Produc; IQR, Interquartile range.

Patient recruitment and organizational problems were identified as main reasons for cost differences, 2) higher actual IMP costs were bearable for most investigators, 3) IMP services for IITs were not a priority for the hospital pharmacy, and 4) closer collaboration between clinical trial unit and hospital pharmacy staff, and sufficient staff for IITs at the hospital pharmacy could improve IMP services.

## Conclusions

Multicenter IITs enrolling patients are particularly at risk for higher IMP costs than planned. These trials are more difficult to plan and logistically challenging, which leads to delays and expiring IMP shelf-lives. IMP services of hospital pharmacies are important for IITs in Switzerland, but need to be further developed.

## Introduction

High quality clinical trials play a key role in the evaluation of therapeutic or preventive health care interventions [1, 2]. However, clinical trials are resource intensive and costly [3, 4]; a situation that has been exacerbated in recent decades by various efforts to improve participant protection and research quality [5, 6]. With limited resources available in academic clinical research, this increasing cost pressure has been particularly difficult for investigator-initiated trials (IITs) [7–10]. Previous research has found that a substantial proportion of IITs are prematurely discontinued due to recruitment, organizational, and financial problems [11], consequently leaving important clinical questions unanswered and resulting in a substantial waste of invested resources.

Various tasks are necessary for the successful conduct of an IIT, all of them generating costs. While some costs are covered through collaborators with regular positions in the hospital, specific trial related costs require third party funding. This can be, but is not limited to salary costs of study staff, fees of ethical or other regulatory authorities, database set-up, data monitoring, laboratory costs, insurance costs etc. They all sum up to the overall costs of a trial [12, 13]. There is little empirical evidence on overall cost structures and individual cost components [12, 13]. In IITs in which an active substance or a placebo is being tested at least as part of an intervention or used as a reference, the development, manufacturing, storage, management, and distribution of the investigational medicinal product (IMP) is a potentially important contributor to the overall costs of the trial [13, 14]. However, a systematic search of the literature did not identify any previous studies that empirically investigated IMP costs in IITs (see S1 and S2 Appendices). The potential differences in the planned and actual costs of IMPs in IITs and the views and experiences of the various stakeholders involved, currently remain unclear. A better understanding of these issues may help to identify risks where IITs involving IMPs could be more costly than planned.

For IITs in Switzerland, hospital pharmacies are typically involved in all services related to IMPs [15]. This study aims to examine (1) the differences in planned and actual costs of IMPs in IITs at a university hospital in Switzerland, and (2) stakeholders´ views and experiences regarding IMP cost differences and hospital pharmacy services for IITs.

## Methods

We chose a working definition of an IMP for our study according to the EU Clinical Trial Regulation referring to a medicinal product that is being tested or used at least as part of an intervention or as a reference (including placebo) in a trial [16]. IMP costs included any of the following: Purchasing price, repacking, labelling, development and manufacturing of one or several batches of IMP, contributing to regulatory approval, storage, logistics, randomisation, and emergency unblinding, if needed.

### Differences in planned and actual costs of IMPs

**Study sample.** All IITs (randomized and non-randomized) that used any service regarding IMPs at the hospital pharmacy of the University Hospital Basel where the hospital pharmacy invoiced its services between January 2014 and June 2020 were included; before 2014 the administrative system of the hospital pharmacy had been re-organized several times.

**Data collection.** A data extraction form was designed using Ninox (https://ninoxdb.de/). We used administrative trial files at the hospital pharmacy that contained (among other documents) trial protocols, descriptions of provided services regarding IMPs, and issued invoices to extract the following information: trial characteristics (number of study centers, study population, planned sample size, trial design, if placebo-controlled, medical field, funding sources, planned and actual trial duration), IMP characteristics (dosage form and production), and IMP costs (planned and actual costs of IMP provision by the hospital pharmacy including, for example, development, preparation, manufacturing, repacking, labelling, randomisation, storage, and fraction of total trial costs). All costs were extracted without considering taxes. The extracted data was pseudonymized and entered into the database by ATH and double-checked by AH; any disagreement was discussed until consensus was reached. If any relevant information was missing from the hospital pharmacy documents, principal investigators were contacted and asked to provide the missing information.

**Data analysis.** Descriptive statistics included absolute and relative frequencies per category and, in order to describe central tendencies and spread, medians, interquartile ranges (25th and 75th percentiles), and ranges were provided. To compare differences between planned and actual IMP costs, the relative change in costs was calculated. The average annual inflation rate has not changed over the last years in Switzerland, and was therefore not considered in the analysis [17]. IITs were classified as "trials with substantial difference" if the difference between planned and actual IMP costs was more than 10% of the planned costs. The chosen cut-off was based on deliberations with involved stakeholders stressing that financial resources for IITs are always scarce and precious. Since most of the included trials were ongoing in 2017, all costs were expressed as mean values of 2017 US$ (1 Swiss Franc = US$ 1.016) [18]. All statistical analyses were conducted using R version 3.5.2. and Excel 2016.

### Stakeholders´ views and experiences

The methods for the interviews are presented in accordance with the "Consolidated criteria for reporting qualitative research" (COREQ) [19] (S3 Appendix).

**Research team and reflexivity.** Interviews were conducted by ATH, a female MSc candidate in Epidemiology with a background in Statistics, and MB, a male physician and Professor in Clinical Epidemiology. SML, a male ethicist, participated in the analysis of the interviews. ATH had no previous experience in qualitative research, while MB and SML both have longstanding expertise in qualitative research. ATH did not have any established relationship with any of the participants, but MB knew most participants from previous work and research. There was no hierarchical relationship between ATH or MB and the interviewees.

**Study design.**    The theoretical framework employed in this study was conventional content analysis [20]. Stakeholders were selected through purposive sampling to ensure participants from the key groups were included. Ten participants agreed to participate in the study and were recruited from three groups: 1) principal investigators whose trials showed a substantial difference between planned and actual IMP costs (n = 5), hospital pharmacists (n = 3), and representatives of the Basel clinical trial unit (CTU) (n = 2). Participants were approached through Email. No stakeholder approached refused to participate. The aim, method, and content of the study was clearly explained to the interviewees. Interviewees were ensured that their participation is voluntary, anonymized and they had the right to withdraw their participation. The oral informed consent was obtained and audio recorded. All interviews were conducted in English and audio recorded, no field notes were taken. Interviews were held between March and May 2020. Three interviews were conducted in person at a venue of the participants' choosing, while the remaining interviews were conducted via video call because of the COVID-19 pandemic restrictions. Only the participant and the researchers were present during the interview. A researcher-developed semi-structured interview guide was developed for each group to guide the discussion (S4 Appendix). Based on the first two interviews that did not show any problems, it was decided that no further piloting or adaptation of the interview guides was necessary. No repeated interviews were carried out. Interviews lasted on average 23 minutes. After 10 interviews, a preliminary analysis was conducted and found no new aspects on the topic emerging, suggesting saturation in the content and attitudes expressed by the participants. Interviews were transcribed verbatim and pseudonymized. Participants have not provided any corrections or feedbacks on transcripts and findings as the interview guide and provided answers were quite simple, short, and unambiguous.

**Qualitative analysis.**    Based on the interview transcriptions, conventional content analysis was performed by ATH using MAXQDA version 11. Initial themes identified common across participants as well as those unique to individuals were labelled using a process of open coding. Findings are presented as higher- and lower-level categories in a coding frame, which was reviewed by MB and SML, and the extracted codes were discussed until consensus was achieved.

## Ethics approval and consent to participate

We presumed that an ethical approval for this study will not be needed, as it is not health-related data and not in the scope of the applicable Human Research Act; however, we submitted a jurisdictional inquiry to the North West and Central Switzerland (EKNZ) for clarification and received a confirmation that further ethical approval was not needed for our study on August 22, 2019. Orally informed consent was obtained from all interviewees in the qualitative part of the study.

## Results

### Differences in planned and actual costs of IMPs

**Study characteristics.**    Overall, 26 IITs were invoiced by the hospital pharmacy in Basel regarding IMP production and pharmacy services between January 2014 and June 2020. Two IITs were excluded because they were discontinued prematurely due to difficulties in patient recruitment. Consequently, 24 IITs were included in the analysis (Table 1). The majority of trials were single center (18/24; 75%), enrolled patients—in contrast to healthy volunteers—(13/24: 54.2%), with a median planned sample size of 45.5 participants, were randomized with a parallel or crossover design (23/24: 95.8%), were placebo-controlled (22/24; 91.7%), in the field of neurology, cognitive neuroscience, or endocrinology (17/24; 70.8%); were not a pilot study

**Table 1. Characteristics of included investigator-initiated trials with and without substantial IMP cost difference\*.**

| | All trials (n = 24) | Trials without substantial cost difference* (n = 13) | Trials with substantial cost difference* (n = 11) |
|---|---|---|---|
| **Trial characteristics** | | | |
| **Center status** | | | |
| Single center | 18 (75.0%) | 13 (100%) | 5 (45.5%) |
| Multicenter, international | 3 (12.5%) | 0 | 3 (27.3%) |
| Multicenter, national | 3 (12.5%) | 0 | 3 (27.3%) |
| **Study population** | | | |
| Healthy volunteers | 11 (45.8%) | 10 (76.9%) | 1 (9.1%) |
| Patients | 13 (54.2%) | 3 (23.1%) | 10 (90.9%) |
| **Planned sample size** | | | |
| Median (IQR) | 45.5 (36–83.3) | 40 (24–46) | 60 (47.5–135) |
| **Trial design** | | | |
| parallel group, non-randomized | 1 (4.2%) | 0 | 1 (9.1%) |
| parallel group, randomized | 12 (50.0%) | 2 (15.4%) | 10 (90.9%) |
| Crossover, randomized | 11 (45.8%) | 11 (84.6%) | 0 |
| **Placebo controlled trial** | | | |
| Yes | 22 (91.7%) | 13 (100%) | 9 (81.8%) |
| No | 2 (8.3%) | 0 | 2 (18.2%) |
| **Medical field** | | | |
| Neurology | 7 (29.2%) | 2 (15.4%) | 5 (45.5%) |
| Cognitive neuroscience | 6 (25.0%) | 5 (38.5%) | 1 (9.1%) |
| Endocrinology | 4 (16.7%) | 2 (15.4%) | 2 (18.2%) |
| Other | 7 (29.1%) | 4 (30.7%) | 3 (27.2%) |
| **Pilot study** | | | |
| Yes | 1 (4.2%) | 0 | 1 (9.1%) |
| No | 23 (95.8%) | 13 (100%) | 10 (90.9%) |
| **Competitive funding from the Swiss National Science Foundation** | | | |
| Yes | 10 (41.7%) | 6 (46.2%) | 4 (36.4%) |
| No | 14 (58.3%) | 7 (53.8%) | 7 (63.6%) |
| **Delayed start of participant enrolment** | | | |
| Yes | 18 (75.0%) | 9 (69.2%) | 9 (81.8%) |
| No | 6 (25.0%) | 4 (30.8%) | 2 (18.2%) |
| **Duration of delayed start** (in months) | | | |
| Median (IQR) | 2 (0.5–7) | 2 (0.5–4.5) | 6.5 (1–11) |
| **Planned trial duration** (in months) | | | |
| Median (IQR) | 14.5 (11.8–24) | 12 (6–15) | 24 (16–32.5) |
| **Actual trial duration** (in months) | | | |
| Median (IQR) | 18 (9.3–30.8) | 10 (5–18) | 28 (19.5–55) |
| **IMP costs** | | | |
| **Planned IMP costs in US$** | | | |
| Median | 11'279 | 11'130 | 11'427 |
| IQR | 8'744–17'392 | 8'882–16'302 | 8'922–36'166 |
| Range | 4'711–85'334 | 6'853–83'718 | 4'711–85'334 |
| **Actual IMP costs in US$** | | | |
| Median | 15'076 | 11'130 | 28'868 |
| IQR | 10'610–28'994 | 8'882–16'302 | 13'004–49'777 |
| Range | 7'072–121'063 | 7'072–69'880 | 9'968–121'063 |

(*Continued*)

**Table 1.** (Continued)

| | All trials (n = 24) | Trials without substantial cost difference* (n = 13) | Trials with substantial cost difference* (n = 11) |
|---|---|---|---|
| **IMP fraction of total trial costs in % ** ** | | | |
| Median (IQR) | 5 (3–12) | 4.8 (2.8–7.3) | 14.5 (3.8–19.5) |
| **Planned IMP costs per patient in US$** | | | |
| Median (IQR) | 254 (116–758) | 312 (208–557) | 130 (65–1'308) |
| **Actual IMP costs per patient in US$** | | | |
| Median (IQR) | 263 (164–763) | 318 (208–557) | 199 (111–2'161) |

Abbreviations: IMP = Investigational Medicinal Product; IQR = interquartile range

* The working definition of a substantial cost difference was that the actual IMP costs were more than 10% higher than the planned IMP costs in a trial.

** Based on available data on overall trial costs; for three trials overall costs were not available and principal investigators then estimated the IMP fraction of total trial costs, and for five trials principal investigators did not respond to provide us with IMP fraction of total trial costs.

(23/24; 95.8%), and had not received support by the Swiss National Science Foundation (14/24; 58.3%). The majority of trials also experienced a delayed start in participant enrolment (18/25; 75%), with a median delayed start of 2 months. Furthermore, trials had an actual median duration of 18 months that was longer than planned (median planned trial duration was 14.5 months).

**IMP characteristics.** As some trials used more than one verum, a total of 29 IMP-verums and 25 IMP-placebos were analyzed (Table 2). The most frequent pharmaceutical dose form for IMPs used by IITs were capsules. IMP-verums were typically produced in two ways: I) The IMP medication with marketing authorization was purchased from a pharmaceutical company and reformulated (encapsulated, blinded) or packed (repacked or relabelled) by the hospital pharmacy (n = 11; 39.3%), or II) an active pharmaceutical ingredient (API) was purchased and the hospital pharmacy did non-sterile manufacturing of the capsules/solution/or another pharmaceutical dose form (n = 10; 34.5%). In the trials using IMP-placebos, matching placebos were most frequently manufactured using excipients only (capsules/solution/other pharmaceutical dose form (n = 14; 56.0%), while in 24% (n = 6) a placebo product was purchased from a pharmaceutical company and the hospital pharmacy reformulated, packed and labelled it to ensure blinding.

**IMP costs.** Overall, the median planned costs for IMPs were 11'279 US$ (Interquartile range [IQR], 8'744–17'392 US$), but the actual median costs incurred were 15'076 US$ (IQR, 10'610–28'994 US$) (Table 1). The IMP costs corresponded to a median of 5% of the total trial costs (IQR, 3%– 12%). In 13 IITs, there was no substantial difference between planned and actual IMP costs (<10%). However, 11 trials had a substantial difference (>10%); with a median of 11'427 US$ (IQR, 8'922–36'166 US$) in planned IMP costs, but a median of 28'868 US$ (IQR, 13'004–49'777 US$) in actual IMP costs. This included two IITs with actual IMP costs that were 180% (18'971 US$) and 400% (24'157 US$) higher than planned (S5 Appendix).

Trials with substantial differences between planned and actual IMP costs more frequently used patients as the study population (10/11; 90.9%); had a higher median planned sample size (60 vs 40 for trials without cost differences), used a randomized parallel group design (10/11; 90.9%), had a longer median delay to start participant enrolment (6.5 months vs 2 months for trials without cost differences), had a longer planned trial duration (median 24 months vs 12 months for trials without cost differences), and had a longer actual trial duration (median 28 months vs 10 months for trials without costs differences) (Table 1). No obvious differences

**Table 2. IMP characteristics.**

| IMP-verum characteristics | Total IMP-verum* (n = 29) | IMP-verum in trials without substantial cost difference** (n = 16) | IMP-verum in trials with substantial cost difference** (n = 13) |
|---|---|---|---|
| **Type of manufacturing the IMP-verum** | | | |
| Bought from company, manufacturing of capsules/ solution/ other pharmaceutical dose form by pharmacy | 7 (24.1%) | 6 (37.5%) | 1 (7.7%) |
| Bought from company, packaging (repackaged, blinded and/ or labelled) by pharmacy | 11 (39.3%) | 4 (25.0%) | 7 (53.8%) |
| Bought from company, labelling by pharmacy (not manufactured) | 1 (3.4%) | 0 | 1 (7.7%) |
| Bought API, manufacturing of capsules/solution/other pharmaceutical dose form by pharmacy (non-sterile) | 10 (34.5%) | 6 (37.5%) | 4 (30.8%) |
| Bought API, manufacturing of infusion/solution for injection/ other pharmaceutical dose form by pharmacy (sterile) | 0 | 0 | 0 |
| **Pharmaceutical dose form of IMP-verum** | | | |
| Capsule | 13 (44.8%) | 10 (62.5%) | 3 (23.1%) |
| Oral solution | 5 (17.2%) | 2 (12.5%) | 3 (23.1%) |
| Sachet | 4 (13.8%) | 2 (12.5%) | 2 (15.4%) |
| Tablet | 4 (13.8%) | 1 (6.3%) | 3 (23.1%) |
| Inhalator | 1 (3.4%) | 0 | 1 (7.7%) |
| Others (e.g. solution/powder for injection) | 2 (6.9%) | 1 (6.3%) | 1 (7.7%) |
| **IMP-placebo characteristics** | Total No. IMP-placebo* (n = 25) | IMP_placebo in trials without substantial cost difference** (n = 15) | IMP_placebo in trials with substantial cost difference** (n = 10) |
| **Type of manufacturing IMP-placebo** | | | |
| Bought from company, manufacturing of capsules/ solution/ other dosage by pharmacy | 4 (16.0%) | 3 (20.0%) | 1 (10.0%) |
| Bought from company, packaging (repackaged, blinded and labelled) by pharmacy | 6 (24.0%) | 2 (13.3%) | 4 (40.0%) |
| Bought from company, labelling by pharmacy (not manufactured) | 0 | 0 | 0 |
| Bought excipient, manufacturing of capsules/ solution/ other dosage form by pharmacy (non-sterile) | 14 (56.0%) | 10 (66.7%) | 4 (40.0%) |
| Bought excipient, manufacturing of infusion/ solution for injection/ other dosage form by pharmacy (sterile) | 1 (4.0%) | 0 | 1 (10.0%) |
| **Pharmaceutical dose form of IMP-placebo** | | | |
| Capsule | 12 (48.0%) | 9 (60.0%) | 3 (30.0%) |
| Oral solution | 4 (16.0%) | 2 (13.3%) | 2 (20.0%) |
| Sachet | 4 (16.0%) | 2 (13.3%) | 2 (20.0%) |
| Tablet | 2 (8.0%) | 0 | 1 (10.0%) |
| Inhalator | 1 (4.0%) | 1 (6.7%) | 1 (10.0%) |
| Others (e.g. solution/powder for injection) | 2 (8.0%) | 1 (6.7%) | 1 (10.0%) |

* There were 5 trials with more than one IMP-verum, 3 trials with more than one IMP-placebo and 2 trials without any IMP-placebo per trial.

** The working definition of a substantial cost difference was that the actual IMP costs were more than 10% higher than the planned IMP costs in a trial.

Abbreviations: API = Active Pharmaceutical Ingredient; IMP = Investigational Medicinal Product

between different types of IMPs were identified in IITs with and without substantial cost difference (Table 2).

To find out more about the reasons for substantial differences between planned and actual IMP costs in the 11 IITs, the hospital pharmacy documents were examined in more detail. The identified reasons are presented in Table 3.

**Table 3. Reasons for substantial differences between planned and actual IMP costs in IITs.**

| Reasons of substantial difference in IMP costs | No. of IITs (total n = 11) |
|---|---|
| Participant recruitment problems leading to longer trial duration and expiration of IMP shelf-lives, i.e. a new batch of IMP (verum and placebo) had to be manufactured resulting in extra costs | 6 |
| Principal investigators decided to increase the original planned sample size, which again resulted in additional manufacturing of IMP batches | 3 |
| Higher logistical costs (more shipments) and higher API costs than planned | 2 |

Abbreviations: API = Active Pharmaceutical Ingredient; IMP = Investigational Medicinal Product;

IIT = Investigator-Initiated Trial

## Stakeholders´ views and experiences

**Reasons for differences between planned and actual IMP costs.** *Variation in budget planning*. It was reported by CTU and hospital pharmacy staff that a key reason for difference between planned and actual costs was investigators often failing to plan or realistically consider IMP costs in budgets; this was often exacerbated by investigators approaching the hospital pharmacy or the CTU at a very late stage of the planning process. Investigators developing trial budgets themselves without the assistance of a CTU or hospital pharmacy staff were most likely to experience cost differences (Table 4).

*Organizational problems*. The vast majority of participants reported that unexpected organizational problems were the main reasons for differences between planned and actual costs of IMPs. Slow recruitment of participants and a delayed start of the trial increased trial duration, causing expiration of IMPs and demanding manufacturing of additional IMP batches. Some participants reported that principal investigators try to save resources by making special deals with pharmaceutical companies or by using leftover medication for their trial. Also, they try to economise by ordering the minimum amount of IMPs necessary (without buffer). In the end, IMPs then needed to be reordered and manufactured in addition to what was planned, leading to higher costs. Other reasons reported by participants included additional shipping costs for the distribution of IMPs to multiple sites, and additional costs to obtain required medication or APIs in good manufacturing practice (GMP) quality (Table 4).

**Impact of higher IMP costs.** *Different views from manageable to prematurely terminating the trial*. The extent to which IITs are affected by cost differences between planned and actual costs were perceived differently across participants. Most principal investigators reported that the absolute cost increase due to IMPs remained manageable and was not critical. However, one investigator and hospital pharmacy staff reported that such differences in IMP costs could constitute a major problem, and provided examples of IITs needing to end recruitment before reaching the planned sample size to avoid the additional costs and time delays involved in manufacturing additional IMP batches (Table 4).

**Hospital pharmacy services for IITs.** *Lack of awareness of services*. Participants reported that trial investigators typically are not aware of available services at the hospital pharmacy, and usually only hear about these services by word of mouth from another colleague or from CTU staff. Hospital pharmacy services regarding IMP manufacturing are not widely advertised, were not seen as a priority for the hospital pharmacy, and there are currently no plans to further develop and increase IMP services for IITs; there is a shortage of staff for these activities and the current workload is only just manageable (Table 5).

*Timeliness of services*. Participants repeatedly mentioned that hospital pharmacy staff endeavoured to fulfil orders in a timely fashion but worked at their limit. Participants from the

**Table 4. Coding scheme regarding different aspects of IMP costs.**

| Theme | | |
|---|---|---|
| **Code** | **Subcode** | **Example Quote** |
| **Reasons for differences between planned and actual IMP costs** | | |
| Variation in budget planning | Failure to include or realistically consider the costs of IMPs by principal investigator | *"...what's not really good calculated or just underestimated is the production of the placebo. Because in general, people think the placebo is nothing only the verum..."*(CTU I) |
| | | *"...So in the beginning, that was clearly our team doing a very naive budget. And now in the last trials, it was the CTU doing all the overview for costs, for monitoring, for on site management, data management, pharmacy, etc. ..."* (PI I) |
| Organizational problems | Slow participant recruitment | *"...The recruiting of the patients is a huge problem, I think. At first when they want to start a study, they think, oh I can recruit e.g. about 100 patients and at the end it's about 10 in this time and I think that's a big problem..."* (HP I) |
| | Trial starts later than planned | *"...you could calculate the study starts on the 10th of January, finally you have some requirements from Swissmedic that the study starts three months later. Of course that is another aspect, because study and medication is actually already available, manufactured, but I don't think that is a main reason. Could be one of the reasons, but I would say more secondary not a primary cost factor..."* (CTU I) |
| | Ingredients not available | *"...it's not possible to just reorder it, sometimes the medicinal product isn't available at the moment ..."* (HP I) |
| | Uncertainty in the planning | *"...But in the beginning mostly you don't know exactly how much, you don't want to have too much. It's also that at the moment you say OK we need I don't know drugs or placebos for twenty more patients..."* (PI I) |
| | | *"...I mean trial site and trial recruitment of course are taking as well a big part to this, right! Because you don't know how good the sites recruit..."* (CTU II) |
| | PIs hastily go for apparent bargains when ordering IMPs themselves | *"...we just received some medication with an expiry date of only three months and patients they get the study medication for 6 months duration, so it's additional cost that patients have to come again. There are I think in general 8 different packages, so we have to focus. It is an escalating dose so we have to focus there as well. And this is of course its double plan as well..."* (CTU I) |
| **Impact of higher IMP costs** | | |
| Different views from manageable to prematurely terminating the trial | Additional manufacturing is expensive | *"...high costs for a new batch of IMP..."* (HP II) |
| | Additional manufacturing is time consuming | *"We probably had a trial recently, and we had to stop it ...I mean what you get the first time, but if you have to go again for the second IMP production, it is expensive and takes a lot of time. So, you need to plan years in advance and you don't know how much you recruit in the last year. So, I would really try to avoid additional IMP production at all cost..."* (PI IV) |
| | PIs find extent of cost differences manageable | *"...I can't remember but I'm talking about 3000 CHF. Maybe it was 5000, but it was a 4 digit number and I had that funding from, you know our clinic had a research fund for internal researcher, it was easy to pay it from there. I think the cost was not the problem..."* (PI II) |
| | HP staff believe cost differences to be problematic for principal investigators | *"...I know that because of slow recruitment and high costs for a new batch of IMP, this trial was terminated early with less patients as planned at the beginning. So, that is one problem that the PI faces and I know others that they have to find different sources for financial input...."* (HP II) |

Abbreviations: CTU = Clinical Trial Unit; HP = Hospital Pharmacy; IMP = Investigational Medicinal Product; PI = Principal Investigator

hospital pharmacy reported that the daily business of supporting patient care is demanding and takes priority. All stakeholders emphasized that contacting the hospital pharmacy early in the planning process of an IIT is crucial. It is important to take into account the time needed for any IMP associated processes (Table 5).

*Satisfaction with services.* In general, trial investigators reported being satisfied with the hospital pharmacy´s collaboration and work regarding IMP manufacturing. Overall, investigators felt that they could always rely on the advice and information provided by hospital pharmacy staff, storage condition of IMPs, randomization procedures, or placebo provision. Hospital

**Table 5. Coding scheme regarding IMP services of the hospital pharmacy.**

| Theme | | |
|---|---|---|
| Codes | Subcodes | Example Quote |
| **Hospital pharmacy services for IITs** | | |
| Lack of awareness of services | Little knowledge about hospital pharmacy services among investigators | *"I think they are really not aware what they could do actually. I think a lot of this potential advertisement within the pharmacy is not adequately used."(CTU II)* |
| | Investigators know from word of mouth | *". . .I have heard from different trial that they use, where they produce placebo for the study. So, I knew that they would be doing these kind of things and then I just ask there. . . ."* (PI II) |
| | Lack of advertisement | *". . .I am not aware of any advertisement for the specific service, I have not heard from other people if they are aware of any overview is really provided. . ."* (HP III) |
| | Not priority of hospital pharmacy | *". . .or this manufacturing of IMPs is not our core business, because our core business is to produce the medication of the patients here at the hospital for routine patients . . ."(HP II)* |
| Timeliness of services | High workload | *". . .I don't think that they do any kind of marketing, let's say. Because I know in general that they are already booked out. . ."* (CTU I) |
| | Staff shortage and limited resources | *". . .in general, if they have everything could work together, so I would say the service that could be provide this due to a personal situation is not really perfect. Because if you have a study medication during the weekend, you couldn't do a study during the weekend because they can't give us the study medication on the weekend,. . ."* (CTU I) |
| | | *". . .six hospital pharmacies around Switzerland. But each of them is under-staffed, each of them complaining about too much work and not interested in having more clinical investigators working with them. . ."* (CTU II) |
| Satisfaction with services | Reliable and competent advice and information | *". . .you know that everything is confirmed. That's I think the advantage, Also the Swissmedic is coming, they as you know that is happening for time to time and then you are happy, that you can good rely on hospital pharmacy. . ."* (PI I) |
| | Randomization | *". . .the other value is that they manage also the randomization. . ."* (PI I) |
| | Storage condition of IMP | *". . . they store the drugs for you"* (PI I) |
| | Placebo provision | *". . .We have to have placebo tablets, but they didn't look alike to the IMP Verum. So they produced these capsules and put the tablets inside. And that was cheap actually. It was a very creative way in finding a solution. . ."* (PI III) |
| | Friendly, dedicated staff | *". . .And I think the pharmacy was great and they try to accommodate the need and we also have all this problems that the company, they don't deliver in the time they should and then the pharmacy has to speed up, so that we still have the medication when we need it. And they were helpful and accommodating and doing whatever they could do to have the stuff at the time we needed it. . ."* (PI II) |
| Recommendations | Improvement in collaboration between CTU+ HP | *". . .I mean I never met them personally, so you talk to them on the phone and we also discuss we should meet. But so far it never happen. So, I think people who should work or need to work together there should be some kind of regular meeting. It does not need to be monthly. . ."* (CTU II) |
| | Service network of HPs in Switzerland needed | *". . .I think for the production purpose, it could, especially, relative small country like Switzerland really be a good idea to kind of centralize this more to one or two pharmacies who say we can do the GMP production for the researchers, we have the staff and they (HPs) could be implemented to do so. . ."* (CTU II) |

Abbreviations: CTU = Clinical Trial Unit; GMP = Good Manufacturing Product; HP = Hospital Pharmacy; IMP = Investigational Medicinal Product; PI = Principal Investigator

pharmacy staff were perceived as competent, friendly, and willing to help despite being overloaded with work (Table 5).

*Recommendations.* Participants suggested a number of system level changes in order to support IITs more efficiently (Table 5). Closer collaboration and regular exchange meetings between CTU staff and hospital pharmacy staff were suggested as facilitators of a more efficient process. In addition, a hospital pharmacy service network on a national level regarding IMP production for IITs with individual pharmacies complementing each other in terms of competencies and capacities could help making better use of scarce resources. Finally, research policy

makers at the hospital need to acknowledge the necessity of professional pharmacy support for IITs with IMPs and to ensure sufficient pharmacy staff and resources for this task.

## Discussion

### Summary of main findings

To our knowledge, this mixed methods study is the first empirical investigation of hospital pharmacy records on IMP costs for IITs. We found that IMP costs constituted, on average, 5% of the total IIT costs. In about half of IITs we identified a substantial difference between planned and actual IMP costs, which seems most common among multicenter IITs enrolling patients (as opposed to healthy volunteers). This was supported by statements from stakeholders revealing that organizational problems including slow patient recruitment typically lead to trial prolongation, expiring shelf-lives of IMPs, and the need for manufacturing additional IMP batches. IMP services for IITs were not a priority for the hospital pharmacy and there is a shortage of staff and resources available for these activities. It was recommended that a closer collaboration between CTU and hospital pharmacy staff and a national pharmacy service network for IITs could improve the provision of IMPs for IITs.

### Comparison with similar studies

There is a lack of previous research on planned and actual costs of IMPs in IITs, to which we could compare our results. However, two case studies reported on IMP costs in IITs without comparing planned and actual IMP costs [13, 14]. One case study consisted of a worked example of detailed costs for a typical IIT funded by the Belgian Healthcare Knowledge Centre (20 sites, 400 patients, total trial duration of 4 years) to illustrate their budget calculation tool; based on their expertise they estimated for purchase of the IMP/placebo, blinding, packaging, labelling, recovery and destruction of unused product an amount of Euro 130'000 (about 150'000 US$) which corresponded to 7.5% of the total IIT costs [13]. The other case study consisted of cost details from a placebo-controlled Swiss IIT evaluating adjunct corticosteroids in patients hospitalized with community-acquired pneumonia (7 sites, 802 patients, total trial duration 4.5 years); the IMP costs for that trial were 11'922 US$ accounting for 3% of the total trial costs [14]. The absolute IMP costs in the Belgian paper are by a factor of 10 higher than the median actual IMP costs in our sample, but the IMP cost proportions of the total costs in both cases are well within the IQR of IMP cost proportions in our sample (3%-12%). A recent meta-research study examined costs of placebos used in IITs [21]; median costs for placebos and packaging were 58'286 US$ (IQR, US$ 2'428 to 160'770), accounting for a median of 10.3% of the total trial budget, which is somewhat higher than the median IMP costs found in our study including both, verum and placebo costs (US$ 15'000; median of 5% of total trial budget). Other studies have discussed various fee structures for pharmacy services related to IMPs all dating back to the 1990s [22–27], or evaluated cost savings with pharmacy managed IMPs in breast cancer trials compared to standard treatment for the healthcare system between 2014 and 2016 in Spain [28]. Previous studies about hospital pharmacy services for clinical trials in general from different parts of the world consisted mainly of surveys among hospital pharmacies or more general discussions about the importance and quality of IMP services of hospital pharmacies [29–35]. Reported findings show many similarities with our results from Basel, e.g. shortage of staff and suggestions for hospital pharmacy networks.

## Implications

Swiss ethics committees in Switzerland approve approximately 300 randomized clinical trials each year [36]. Since 2015, the Swiss National Science Foundation specifically supports IITs with 10 Mio CHF per year [37], and thousands of patients make efforts and take risks in participating in these trials. Therefore, the need to ensure that we apply suitable and evidence-based methods to make clinical trials efficient and sustainable is crucial. Insights from our project provide trial investigators and other stakeholders with examples of empirical data on IMP costs in IITs and underline the importance of careful trial planning together with trial support units [11, 38, 39]. The role and value of hospital pharmacies in IITs with IMPs needs to be recognized by all clinical trial stakeholders in Switzerland and a collaborative effort across stakeholders is necessary to remediate existing inefficiencies and resource limitations. Further empirical evidence on IMP services for IITs outside of Switzerland would be relevant to allow for comparisons of research infrastructure across countries in order to identify most promising approaches.

## Strengths and limitations

Strengths of our study include the fact that we had full access to all documents of eligible IITs at the hospital pharmacy and all extracted data were double-checked by a staff member of the hospital pharmacy in order to minimize any potential extraction errors. All principal investigators of included IITs were responsive upon email contact and provided additional information about their trials where needed. For the qualitative part, we used purposive sampling and managed to interview representatives of important stakeholder groups allowing for a broad spectrum of viewpoints. Open-ended questions gave our respondents the freedom to openly discuss their ideas allowing for a rich qualitative analysis. Finally, we qualitatively analyzed interview transcripts in a team of three researchers minimizing biased interpretation of interview statements.

This study has several limitations: First, we investigated IMP costs and hospital pharmacy services at one university hospital in Switzerland. Although stakeholders reported in interviews that according to their knowledge the situation is similar at other Swiss university hospitals, the extent to which our findings are generalizable nationally and internationally remains to be determined. Second, our chosen cut-off of more than 10% difference between planned an actual IMP costs to be considered substantial could be perceived as arbitrary and too low rendering our results little robust. However, as shown in S5 Appendix a higher cut-off of e.g. 20% would be unlikely to change our results since only one trial would switch categories from substantial difference to no substantial difference. Third, due to the small sample size, we did not perform any multivariable regression analysis but only summarized absolute and relative numbers descriptively with one stratification variable (presence of a substantial difference between planned and actual IMP costs). Fourth, we included all IITs, which were invoiced after January 2014, which means that they could have started long before; the earliest included trial actually started in 2007. Some processes, regulatory requirements or legislation might have changed during this time period. Fifth, we could not distinguish between the costs of placebo and verum in this study due to administrative reasons at the hospital pharmacy. According to hospital pharmacy staff, potential differences between verum and placebo manufacturing costs are negligible in most cases as the costs for the API contribute only to a small extent to the overall price of the manufacturing of a batch of IMP; the more relevant point is, whether and how IMP verum and IMP placebo are available from a pharmaceutical company. At pharmaceutical companies, verum which often refers to a marketed drug is typically manufactured on large scale batches and, therefore, is considerably cheaper and easily available compared to a

matching placebo. Only few pharmaceutical companies are willing or in a position to manufacture matching placebos on demand. In addition, pharmaceutical companies appear more willing to provide the placebo if a clinical trial is within their interest [21].

## Conclusions

Multicenter IITs enrolling patients are particularly at risk for higher IMP costs than planned. These trials are more difficult to plan and logistically more challenging than single center trials or trials with healthy volunteers. This can lead to trial prolongation and expiring shelf-lives of IMPs and the need for additional IMP manufacturing. Involved stakeholders agreed that IMP services for IITs are not a priority for hospital pharmacies in Switzerland. Investigators should contact the hospital pharmacy early in the planning process for an IIT. In addition, closer collaboration between the CTU and the hospital pharmacy could make the planning of IITs for investigators easier and more efficient, sufficient resources and staff at hospital pharmacies to support IITs, and a national service network of hospital pharmacies would be a way forward to improve the infrastructure for IITs in Switzerland.

## Supporting information

**S1 Appendix. Search strategy.**
(DOCX)

**S2 Appendix. PRISMA flow diagram.**
(DOCX)

**S3 Appendix. COREQ checklist.**
(DOCX)

**S4 Appendix. Interview guide.**
(DOCX)

**S5 Appendix. Figure on relative increase in planned vs. actual costs of IMPs.**
(DOCX)

## Acknowledgments

The authors thank Amanda Herbrand for her assistance with Ninox, which we used for data collection.

MARTA (MAking Randomized Trials Affordable) group members (group leader is marked with *):

**Benjamin Speich**\* (Benjamin.speich@usb.ch), Basel Institute for Clinical Epidemiology and Biostatistics, Department of Clinical Research, University of Basel and University Hospital Basel, Basel, Switzerland / Oxford Clinical Trials Research Unit, Centre for Statistics in Medicine, Nuffield Department of Orthopaedics, Rheumatology and Musculoskeletal Sciences, University of Oxford, Oxford, UK **Lars G. Hemkens**, Basel Institute for Clinical Epidemiology and Biostatistics, Department of Clinical Research, University of Basel and University Hospital Basel, Basel, Switzerland

**Alain Amstutz**, Swiss Tropical and Public Health Institute, and Division of Infectious Disease and Hospital Epidemiology, University of Basel and University Hospital of Basel

**Benjamin Kasenda**, Basel Institute for Clinical Epidemiology and Biostatistics, Department of Clinical Research, University of Basel and University Hospital Basel, Basel, Switzerland

**Christiane Pauli-Magnus**, Clinical Trial Unit, Department of Clinical Research, University of Basel and University Hospital Basel, Basel, Switzerland

**Matthias Briel**, Basel Institute for Clinical Epidemiology and Biostatistics, Department of Clinical Research, University of Basel and University Hospital Basel, Basel, Switzerland

**Matthias Schwenkglenks**, Institute of Pharmaceutical Medicine, University of Basel, Basel, Switzerland

**Alexandra Griessbach**, Basel Institute for Clinical Epidemiology and Biostatistics, Department of Clinical Research, University of Basel and University Hospital Basel, Basel, Switzerland

**Ala Taji Heravi**, Basel Institute for Clinical Epidemiology and Biostatistics, Department of Clinical Research, University of Basel and University Hospital Basel, Basel, Switzerland

**Stuart McLennan**, Institute of History and Ethics in Medicine, Technical University of Munich, Munich, Germany

## Author Contributions

**Conceptualization:** Ala Taji Heravi, Vera Ruth Mitter, Matthias Briel.

**Data curation:** Ala Taji Heravi, Anne Henn, Stefanie Deuster, Stuart McLennan, Viktoria Gloy, Matthias Briel.

**Formal analysis:** Ala Taji Heravi, Stuart McLennan, Matthias Briel.

**Investigation:** Ala Taji Heravi, Anne Henn, Stefanie Deuster, Stuart McLennan, Viktoria Gloy, Matthias Briel.

**Methodology:** Ala Taji Heravi, Anne Henn, Stefanie Deuster, Vera Ruth Mitter, Matthias Briel.

**Resources:** Ala Taji Heravi.

**Supervision:** Matthias Briel.

**Visualization:** Ala Taji Heravi, Stuart McLennan, Matthias Briel.

**Writing – original draft:** Ala Taji Heravi, Matthias Briel.

**Writing – review & editing:** Ala Taji Heravi, Anne Henn, Stefanie Deuster, Stuart McLennan, Viktoria Gloy, Vera Ruth Mitter, Matthias Briel.

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
