## [Decision Letter · Decision Letter 0]

15 Dec 2021

PONE-D-21-09088Investigational Medicinal Products, related Costs and Hospital Pharmacy Services for Investigator-Initiated Trials: An empirical mixed-methods studyPLOS ONE

Dear Dr. Taji Heravi,

Thank you for submitting your manuscript to PLOS ONE. After careful consideration, we feel that it has merit but does not fully meet PLOS ONE’s publication criteria as it currently stands. Therefore, we invite you to submit a revised version of the manuscript that addresses the points raised during the review process.

We look forward to receiving your revised manuscript.

Kind regards,

Ismaeel Yunusa, PharmD, PhD

Academic Editor

PLOS ONE

Journal Requirements:

2. Thank you for stating the following financial disclosure: "There were no specific funds available for this project."

3. Thank you for stating the following in your Competing Interests section: "All authors declare no financial relationships with any organization that might have an interest in the submitted work and no other relationships or activities that could appear to have influenced the submitted work."

5. Please upload a copy of Supporting Information S1-S5 Appendix which you refer to in your text on page 34.

Reviewers' comments:

Reviewer's Responses to Questions

**Comments to the Author**

1. Is the manuscript technically sound, and do the data support the conclusions?

Reviewer #1: Yes

Reviewer #2: Partly

Reviewer #3: Yes

2. Has the statistical analysis been performed appropriately and rigorously? 

Reviewer #1: Yes

Reviewer #2: Yes

Reviewer #3: Yes

3. Have the authors made all data underlying the findings in their manuscript fully available?

Reviewer #1: Yes

Reviewer #2: No

Reviewer #3: Yes

4. Is the manuscript presented in an intelligible fashion and written in standard English?

Reviewer #1: Yes

Reviewer #2: Yes

Reviewer #3: Yes

5. Review Comments to the Author

Reviewer #1: This paper is an interesting paper and it brings to light challenges experienced in conducting clinical trials.

Here are some of the comments for the authors to take note of:

1. Last paragraph in the Quantitative results, where the authors discussed on the reason for cost differences, I would suggest they present it in tabular form for easier reading as it passes an important information.

2. In the discussion, the summary of findings seems quite long and was a repetition of what was already mentioned in the Results section. I would suggest the authors discuss their results together with the comparison with other studies (although the mentioned few studies available for comparison) to prevent multiple repetitions

3. Line 292 replace higher than ‘then’ to higher than ‘the’

Reviewer #2: Overall, this is an interesting and well written manuscript that picks up on a completly unstudied area that deserves furthes investigation. More details should be provided in the background section. The qualitative part of the study is quite strong. I have some observations to the quantative part of the study. I also wonder this is in fact a mixed methods design. There is a qualitative and a quantitative part, while I wonder whether the connection between them is strong or whether both parts could had also been published as separate parts.

I have the following comments:

- consider whether the title is appropriate. What would be a non-empirical mixed-methods-study? Aren´t they all empirical?

- as the cut-off of 10% is crucial for your analysis its definition must be placed in the abstract

- The introduction would benefit from more background information as this topic will be rather new for most readers. - With IMP you focus on one potential contributor to the costs. What are the other contributors?

- line 63: 'In IICTs testing drug substances' ... I am not sure what this does mean? Does this refer to pharmacological trials (drug substance is at least part of an intervention)? For my understanding, IMP costs can also occur in a surgical trial. Please clarify your concepts and definitions. In particular clearly define IMP.

- Following this, I fail to understand whether your included trials tested drug substances?

- line 79: does the time span relate to the duration of the trials or to receiving the invoice?

- I have no idea what is meant by hospital pharmacy documents. Please explain more in detail. It is important for the reader to gain sufficient knowledge of the data source. I think that this information can be easily provided in the supplement.

- line 117: add 'were'

- why did you only extract whether there was 'Swiss National Science Foundation support'? What about other sources of funding? There might also be internal hospital funding, for example.

- was there a protocol for your study? please state. This needs to be state as the cut-off of 10% additional costs looks a bit arbitrary to me. From my perspective, 10% is not much and is something that can be expected. Is there any rationale for the 10%? Otherwise you should consider whether a sensitivity analysis might be useful. It definitely needs to be discussed as you had a special focus on principal investigators where the trial had a substantial difference. If the cut-off would have been set higher (eg. 20%) you might not have been able to recruit investigators for your interviews. Please provide more details on the actual absolute difference in costs (median, IQR, range).

- I wonder whether the results in table 2 can be described in a more informative way. You have data on IMP-verum nested in trials. Your n is quite los, so I am not sure that this can be considered at all. However, as a minimal solution I would suggest to describe in how many studies more than one IMP-varum was used.

- the qualitative part of the study is well described and informative

- the same is true for the discussion

Reviewer #3: Thank you for giving me the opportunity to review this interesting manuscript. I have only some minor suggestions for improvement.

- Can you give more detail what the costs of IMP include and how these were determined? Are these only the purchasing prices? As the pharmacy service was also involved in preparing (e.g. packing) the IMP this would that personal costs and thus a large share of costs were not considered.

- I would suggest to report the average cost per patient instead of absolute costs to improve comparability and interpretation in general.

- Are the reported cost the total cost in a certain trial?

- Were data on costs missing? If so, it would be good to have more information on missing cost data, considering that if principle investigators guessed this it is probably imprecise.

6. PLOS authors have the option to publish the peer review history of their article (what does this mean?). If published, this will include your full peer review and any attached files.

Reviewer #1: **Yes: **Dr. Basira Kankia Lawal

Reviewer #2: No

Reviewer #3: **Yes: **Tim Mathes

---

## [Author Response · Author response to Decision Letter 0]

4 Feb 2022

Basel, February 4, 22

Ref. ATH

MS ID#: PONE-D-21-09088

First revision: Investigational medicinal products, related costs and hospital pharmacy services for investigator-initiated trials: A mixed-methods study

Dear Dr. Yunusa,

We thank you for reviewing our manuscript and for considering a revised version for publication in PloS One. We have addressed all of the points raised by you and the reviewers and we are grateful for the constructive comments that helped to further improve our manuscript.

Please consider the following Editor’s specific comments (No. 2 and 3) in order to change in online form on our behalf.

• The authors received no specific funding for this project. VRM is supported by the Swiss National Science Foundation through an Early Postdoc.Mobility career grant (P2BEP3_191798).

• The authors have declared that no competing interests exist.

All contributors have approved this revised version of the manuscript and fulfill criteria for authorship.

We hope that the revised manuscript meets your and the reviewers’ expectations and that it can be accepted for publication in the present form. The point by point response is already uploaded.

Sincerely yours,

Ala Taji Heravi

Basel Institute for Clinical Epidemiology and Biostatistics, Department of Clinical Research

University Hospital Basel, Basel, Switzerland; Email: ala.tajiheravi@usb.ch

---

## [Editor Report · Decision Letter 1]

11 Feb 2022

Investigational Medicinal Products, related Costs and Hospital Pharmacy Services for Investigator-Initiated Trials: A mixed-methods study

PONE-D-21-09088R1

Dear Dr. Taji Heravi,

We’re pleased to inform you that your manuscript has been judged scientifically suitable for publication and will be formally accepted for publication once it meets all outstanding technical requirements.

Kind regards,

Ismaeel Yunusa, PharmD, PhD

Academic Editor

PLOS ONE
---

## [Editor Report · Acceptance letter]

24 Feb 2022

PONE-D-21-09088R1 

Investigational medicinal products, related costs and hospital pharmacy services for investigator-initiated trials: a mixed-methods study 

Dear Dr. Taji Heravi:

I'm pleased to inform you that your manuscript has been deemed suitable for publication in PLOS ONE. Congratulations! Your manuscript is now with our production department. 

Kind regards, 

on behalf of

Dr. Ismaeel Yunusa 

Academic Editor

PLOS ONE